# Functional Analysis Validation of Micro and Conventional Injection Molding Machines Performances Based on Process Precision and Accuracy for Micro Manufacturing

**DOI:** 10.3390/mi11121115

**Published:** 2020-12-16

**Authors:** Matteo Calaon, Federico Baruffi, Gualtiero Fantoni, Ilenia Cirri, Marco Santochi, Hans Nørgaard Hansen, Guido Tosello

**Affiliations:** 1Department of Mechanical Engineering, Technical University of Denmark, Building 427A, Produktionstorvet, DK-2800 Kgs Lyngby, Denmark; federico.baruffi.91@gmail.com (F.B.); hnha@mek.dtu.dk (H.N.H.); guto@mek.dtu.dk (G.T.); 2Department of Civil and Industrial Engineering, University of Pisa, Largo Lucio Lazzarino 2, 56126 Pisa, Italy; g.fantoni@ing.unipi.it (G.F.); marco.santochi@unipi.it (M.S.); 3Erre Quadro s.r.l., Largo Padre Renzo Spadoni c/o Cittadella Galileiana, 56126 Pisa, Italy; ilenia.cirri@gmail.com

**Keywords:** functional analysis, micro-injection moulding, machine design, process capability

## Abstract

Micro polymer parts can be usually manufactured either by conventional injection moulding (IM) or by micro-injection moulding (µIM). In this paper, functional analysis was used as a tool to investigate the performances of IM and µIM used to manufacture the selected industrial component. The methodology decomposed the production cycle phases of the two processes and attributed functions to parts features of the two investigated machines. The output of the analysis was aimed to determine casual chains leading to the final outcome of the process. Experimental validation of the functional analysis was carried out moulding the same micro medical part in thermoplastic elastomer (TPE) material using the two processes by means of multi-cavity moulds. The produced batches were assessed using a precision scale and a high accuracy optical instrument. The measurement results were compared using capability indexes. The data-driven comparison identified and quantified the correlations between machine design and part quality, demonstrating that the µIM machine technology better meets the accuracy and precision requirements typical of micro manufacturing productions.

## 1. Introduction

Conventional injection moulding (IM) is the most used process for the manufacturing of polymer parts, since it enables the mass-production of net-shaped components. In the last decades, the miniaturization of components has become one of the principal technological drivers in many engineering sectors [1]. In order to meet the consequent growing demand of micro components, conventional injection moulding (IM) was downscaled into micro-injection moulding (µIM) [2]. The two technologies have the same process cycle phases (i.e., filling, packing, cooling, demoulding) but, at the same time, fundamental differences deriving from the smaller dimensional scale exist between the process chains associated with both processes. In particular, specific micro tooling processes [3,4,5], micro scale measuring techniques [6] and new design approaches [7,8] must be adopted when dealing with micro scale polymer processing [9,10]. New injection moulding machines have been also developed: conventional ones embed a reciprocating screw, while those dedicated to µIM typically have a screw for plasticising pellets and a separate plunger (diameter of 5 mm down to 2/mm) for metering and injection [11]. Such alternative architecture increases the accuracy of polymer melt dosing and provides higher injection speeds because of the lighter and more controllable injection plungers. This directly results in higher repeatability and improved replication fidelity. These features make µIM the preferred method for the manufacturing of polymer micro parts [12]. However, it is worth noticing that polymer micro parts can be manufactured by both IM and µIM. IM is usually employed in the plastic industry when small batches of micro components are needed and, therefore, the investment related to the acquisition of a dedicated µIM machine is not sustainable or justified.

Although the differences between using an IM and a µIM machine are well known, no study reports an investigation aimed at correlating the functionality of the two machine architectures to the actual dimensional capabilities of both the macro and micro process. In fact, the literature mostly focused on the difference in terms of morphology, demonstrating that the size of crystalline entities was significantly influenced by the dimensional scale of the moulded component [13,14].

In order to investigate the impact of a particular machine layout on its function, functional analysis and axiomatic design represent powerful tools. Functional Analysis (FA) allows identifying the functions performed by the product and by the components of the structure that carry on these functions [15,16,17,18]. It has already been used in the field of injection moulding by the authors [19] to analyse two different machine designs. Axiomatic Design, correlating functional requirements with parameters, allows a comparison between different design arrangements [20]. Functional Requirements are defined as the “minimum set of independent requirements that completely characterize the functional needs of the product in the functional domain”. The fundamental axiomatic axioms can be read together as “Among all the design that satisfy the independence axiom, the one with the minimum information content is the best design”. It means that Design Parameters, which are the physical variables characterizing the physical entities, are related to Functional Requirements in such a way that specific parameters can be adjusted to satisfy its corresponding requirements without affecting the others. Since axiomatic design links functions with corresponding features, it is highly relevant for both designers and production engineers.

In this paper, an IM and a µIM moulding machine were directly compared by using FA and then axiomatic design in order to demonstrate the differences in terms of process capabilities.

The relevant machine macro-functions, corresponding to the main phases of the moulding process, were used to divide the conventional and the micro process in functional maps. The map differences represented the functional key aspects, allowing to identify critical components and possible working problems.

The axiomatic analysis began with an identification of the design key features that affect the feeding phase (therefore the metering) and the injection phase of the process. Those two phases are the most different in the implementation and the most critical ones. For such a reason, these two were firstly compared in an aggregate axiomatic analysis and then the entire axiomatic matrix of each machine was built.

The same micro part was moulded with the two machines and dimensionally assessed. Data on precision (i.e., repeatability) and accuracy (i.e., closeness to target, that in the context of the moulding processes considered in this research is represented by the cavity dimensions) of IM and µIM were gathered in order to validate the functional and axiomatic analysis results.

## 2. Materials and Methods

### 2.1. Case Study

The investigated micro part was a thermoplastic elastomer (TPE) component for medical applications with a nominal mass of 20 mg. TPE was selected for its elastic properties as well as a level of mouldability that enabled an effective and repeatable micro replication process [21]. Figure 1 shows the geometry of the micro part, which is cylindrical and has a through hole generated by a pin coaxial to the cavity. The main dimensional features of the part are shown, namely inner and outer top diameters (IDt and ODt), inner and outer bottom diameter (IDb and ODb), and two lengths (L1 and L2). In particular, IDb and ODt were chosen as indicators for comparison of µIM and IM since they are geometries originated by the replication of the cavity wall and the pin. There exists a significant distinction between these two situations, since the polymer is allowed to shrink freely in correspondence with the outer diameter, while it undergoes a constrained shrinkage when inner dimeters are involved, thus generating residual stresses that enhance the deviation with respect to the mould dimensions. The other geometries were also assessed in order to determine the volume of each moulded part, which was then employed for the density calculation.

The density is a particularly relevant output of any moulding process since it is an indication of the holding phase performance. A higher resulting density means that the packing phase was particularly effective in achieving a high shrinkage compensation. Compensating for shrinkage allows to obtain a higher replication degree of the moulded part with respect to the cavity geometry, higher dimensional accuracy and lower warpage (i.e., smaller form errors).

The dimensional tolerances were specified as ± 50 µm on the considered geometries. The polymer used for both IM and µIM experiments was a Thermolast^®^ grade from Kraiburg TPE GmbH (Waldkraiburg, Germany) having a nominal density of 0.89 g/cm^3^. The viscosity and pressure-specific volume-temperature plots of the material are presented in Figure 2.

### 2.2. IM and µIM Set-Ups

IM experiments were performed using an Allrounder 270 U injection moulding machine from Arburg (Lossburg, Germany) equipped with an 18 mm diameter reciprocating screw and capable of a maximum clamping force of 400 kN. A two-plate mould with four cavities was used (see part, gate and runner system layout in Figure 3a). The volume of the feed system was equal to 980 mm^3^, accounting for 91% of the total amount of injected polymer. The four injection moulded parts account for a total of 96.8 mm^3^ (the nominal volume of one part based on design specifications is 24.2 mm^3^), equal to 9.0% of the total injection volume. The usage of submarine pin gates allowed achieving automatic detachment of the parts from the feed system [22].

µIM experiments were carried out with a state-of-the-art MicroPower 15 micro injection moulding machine from Wittmann-Battenfeld (Vienna, Austria). This machine presents a 14 mm diameter plasticisation screw and a 5 mm injection plunger. The maximum clamping force is equal to 150 kN. A two-plate micro injection moulding tool with four cavities was used with this machine (see part, gate and runner system layout in Figure 3b). The feed system was designed with a submarine gate and had a total volume of 174 mm^3^, thus representing 64.3% of the total injected shot. Correspondingly, the four injection moulded parts, which account for a total of 96.8 mm^3^, represent 35.7% of the total micro injection volume. By comparing this value with the one of the previous case, it is clear that µIM allowed to consistently reduce the amount of material waste, representing a valuable improvement with respect to production cost reduction, material consumption and production sustainability.

Table 1 shows the optimized settings for the two processes. The same level of holding pressure, melt temperature and mould temperature were kept in order to minimize the sources of variation in the comparison between IM and µIM. As for the other process parameters, a higher value of injection speed was used with the µIM machine in order to balance for the smaller injection section. µIM was set on a shorter cycle time due to the smaller amount of injected polymer into the cavity.

### 2.3. Measurement Strategy and Uncertainty Evaluation

After discarding the first 50 shots, 10 consecutively injected parts were collected per each of the four mould cavities and then weighed for both IM and µIM batches using a scale having 0.1 mg resolution (AW220, Shimadzu Corp., Kyoto, Japan). The 80 moulded micro components were also dimensionally assessed. In particular, the diameters were measured with a 3D focus variation microscope (Alicona InfiniteFocus, Alicona Imaging GmbH, Raaba, Austria) with a 5× magnification objective (0.41 µm vertical resolution and 1.75 µm lateral digital resolution). To do this, top and bottom sides of each part were acquired and then levelled by applying a planar correction to correct for any influence of tilting. After this operation, the measurands were extracted by fitting the points of the measured circles (see Figure 4) with the software MountainsMap^®^ (Digital Surf, Besan çon, France). Each acquisition was repeated three times. The two lengths L1 and L2 were measured with an optical CMM (DeMeet 220, Schut Geometrical Metrology, Groningen, The Netherlands) having a 0.5 µm resolution.

Based on the measurements of the six dimensions (four diameters: ODt, IDt, ODb, IDB; two length: L1, L2), the volume V of each moulded part was calculated as follows:(1)$$V\;=\;\pi \frac{{\mathrm{ODt}}^{2}\;{-\;\mathrm{IDt}}^{2}}{4}\left(L1\;-\;L2\right)+\pi \frac{{\mathrm{ODb}}^{2}\;{-\;\mathrm{IDb}}^{2}}{4}\left(L2\right)$$

The density was then calculated as the ratio of mass and volume.

The cavities of both IM and µIM moulds were measured with an optical microscope having a 2.6 µm lateral resolution (Infinity X-32, DeltaPix, Smørum, Denmark). In particular, the geometries correspondent to ODt and IDb were measured in order to calibrate the process for the comparison of the achieved precision and accuracy. Any influence induced by differences of the mould dimensions between the conventional injection moulding tool and the micro injection moulding tool was then eliminated from the process analysis.

The measurement uncertainty U was evaluated by applying the method described in ISO 15530-3 [24]. This evaluation technique is based on the substitution method, which allows estimating the error of the measuring instrument by repeated measurements on a calibrated artefact that is similar to the actual measurand. Two calibrated artefacts were used: a calibrated circle for the focus variation measurement and calibrated lines for the optical CMM measurements. Four uncertainty contributions were taken into account: u_cal_, as the uncertainty of the calibrated artefacts; u_p_, introduced by the measurement procedure and calculated as standard deviation of 20 repeated measurements on the artefact; u_w_, associated with material and manufacturing variations of the actual measurand; and ures, introduced by the limited resolutions of the instrument. u_w_ was calculated as:(2)$$u_{w}\;=\;\frac{\max\left(M\right)-\;\min(M)}{2\sqrt{3}}$$
where M is the vector listing the three repeated measurements for any of the six measurands. The three contributions were then combined using the law of propagation of uncertainty to determine the expanded uncertainty U:(3)$$U\;=\;k\;\cdot {\left(u^{2}{}_{\mathrm{cal}}{+u}^{2}{}_{p}{+u}^{2}{}_{w}{+u}_{\mathrm{res}}{}^{2}\right)}^{1/2}$$
where k is the coverage factor of 2 selected to achieve a 95% approximated confidence. Table 2 and Table 3 show the uncertainty budgets for IM and µIM parts respectively. Considering the 50 µm tolerance, uncertainty-to-tolerance ratios ranging between 3% and 5% were attained, thus confirming that the employed measuring instruments were suitable for the task [25]. By applying the rule of propagation of uncertainty [26] to the volume and consequently the density formulas, the expanded uncertainty for the density was calculated. In particular, the expanded uncertainty U for the density was on average equal to 0.0014 g/cm^3^ and 0.0017 g/cm^3^ for IM and µIM parts respectively. Such values are much lower than the nominal density: they represent 0.16% and 0.19% of the density of the moulding material in case of IM and µIM respectively. This result confirms that the selected measurement chain was capable of providing a sufficiently accurate output.

## 3. Results

### 3.1. Functional and Axiomatic Results

A machine performance is influenced by its design. The combined application of functional analysis and axiomatic design, as it is done below, allowed to compare the two different moulding processes. The methodology can be extended to a multiple number of machines, which is not the focus of the present study. In the following analyses, Arburg machine architecture and the Battenfeld patent by Ganz [27] were used as the principal source of information with regards to machine designs. Design differences, highlighted by the following functional and axiomatic considerations, refer to the following different machine assemblies (see Figure 5).

The functional analysis of the two machines was organized following the main phases of the moulding process reported below (see Figure 6). Such phases were used to divide the conventional and the micro process in functional maps.

The plastication phases for the IM and µIM machines are reported in Figure 7 and Figure 8 respectively. The screw in both machines carries out the plastication phase. In the functional analysis, the plastication is divided into the solid sub-phase and the liquid sub-phase corresponding to the two different physical states of the material during the process step. The functional maps are similar, but there are two important differences:
The friction generated by the IM machine is greater, since the IM machine screw has a bigger diameter than the µIM machine. Moreover, it is also heavier.The last functional block of the phase, referred to as “store”, takes place in front of the screw in the IM machine, while in the µIM one, the liquid is stored at the beginning of the bore hole (indicated as 9 in previous Figure 5).

The second phase (feeding) is carried out by the bored hole in the µIM machine and again by the screw in the conventional one. In this way, the IM machine sticks to the previous functional map: the injection chamber is fed by rotating the screw. Therefore, there was no need to generate a new functional map, as the feeding is performed simultaneously with the plastication. On the other hand, with the µIM machine the feeding phase is carried out through the bored hole that guides the mould material and controls the mould volume using the pressure sensor situated in the hole (bored hole and pressure sensor are respectively indicated as 9 and 10 in Figure 4). The functional map for µIM machine feeding phase is reported in Figure 9 below.

The last two phases (injection and packing) are performed by the screw in the IM machine (Figure 10 and Figure 11) and by the plunger in the µIM machine (Figure 12). In the IM machine, the screw stops rotating and begins to accelerate and, at the same time, begins the injection in the mould. On the other hand, the µIM machine has a rapid sequence of injection and packing made by the plunger, which contacts the liquid material when its acceleration has already begun (in the IM machine, in fact, the screw begins its acceleration when the liquid material is already accumulated ahead). The functional analysis gives evidence that:The sealing function provided by the plunger and the screw (and consequently the different backflow pressure) are very different during the injection phase. Almost no backflow is observed with the µIM machine if compared to the IM machine. This is due to the plunger’s smaller diameter (so to its tighter tolerance) and to the “sealing effect” realized by solidified material (remaining from previous injections) close to the front of the injection plunger.The effect of air and pneumatic energy that comes out of the mould during the injection phase is different for the two machines (as it highlighted in bold block in Figure 11). To accelerate the screw in IM, a certain stroke is necessary, thus it implies a certain volume of air in front of the screw. Conversely, owing to a smaller mass, a shorter stroke is requested to reach the same speed, and the smaller diameter implies a reduced air storage.

Based on such aspects, it is possible to conclude that the critical aspects (friction, backflow and pneumatic air) have a different impact on the two machines. In particular, they are more relevant in the IM machine, thus decreasing its controllability and consequently its performance.

Considering the two most critical process phases (metering and injection) in the two machines, an axiomatic comparison between the two machines can highlight how their design differences can affect their performance. The screw, in fact, in both machines carries out the metering of liquid material during the feeding phase. However, this component is different for the two machines.

The screw of the IM machine has a greater diameter (D_screw_) and a different shape: Actually the metering section is longer in the µIM, thus assuring an improved control over the feed liquid. The diameter of the µIM being smaller, a more accurate tolerance (T_screw_) than the screw of the IM machine (d_screw_, t_screw_, l_screw_) can also be obtained.

Besides the benefit due to the previous design conditions, the µIM machine also presents an additional key component for guaranteeing a high quality metering: a pressure sensor in the bored hole that accurately measures the liquid material characteristic in front of the injection plunger. Such differences are represented in Table 4, which is arranged into three blocks (a red block, a green block and a yellow block) that influence the performance of the machines. The red block has an impact on the control of the liquid volume, while the green block affects the liquid’s backflow. The yellow block is present only in the µIM machine and its function is to measure the metered volume of the polymer. The more accurate tolerance of the diameter and the presence of the yellow block increase the precision of the µIM machine. The less accurate screw facilitates backflow, while the absence of a proper measuring system hinders a precise metering of the molten plastic.

The injection phase is performed by the screw in the IM machine and by the plunger in the µIM one. The plunger (l_plunger_) is shorter than the screw (L_screw_) and it has a smaller diameter (d_plunger_ and t_plunger_). The µIM machine is also equipped with a strain gauge sensor on the back of the injection plunger [14]. Such different features are listed in Table 5. The main implication is on the two different masses of the screw (red block) and the plunger (green block). Such a difference implies the different possibility of accelerating and decelerating, respectively, the piston and the screw. The piston is lighter than the screw, so it can be rapidly accelerated and decelerated even if it reaches high injection speed values. Another important difference is that in front of the screw there is already liquid material when it is accelerated to reach the desired injection speed. The screw acceleration and the liquid injection are simultaneous events in the conventional machine. Moreover, when the screw accelerates it might also drag plastic pellets and molten liquid.

In view of the aforementioned, the valve in the screwed micro machine does not have rigorous on-off states caused by a difficult control on screw behaviour, while the plunger for its features (smaller mass and sensor presence) can be totally controlled. Furthermore, the shape of the IM screw (see Figure 13) negatively influences the control of its on-off behaviour, in contrast, the plunger cylindrical form does not cause the same effect. In this way, the plunger can act in the µIM machine as a perfect valve so that both the closing and the opening functions are executed with higher precision.

Considering both tables (Table 4 and Table 5), it is possible to observe that in the IM machine the listed features are the same for injection and metering, while in the µIM machine there are other features involved in the two phases.

Extending previous axiomatic considerations in order to build the typical axiomatic matrix, it is possible to observe how principal machine functional requirements (matrix rows) match with their design parameters (matrix columns).

In the following, Table 6 and Table 7 describe the design parameters of the IM and µIM machines respectively. Table 8 and Table 9 present the axiomatic matrixes in which the machine design parameters are used. The two machine functional requirements are referred to previous functional analysis and main phases.

In particular, the screw diameter parameters are omitted in the µIM machine because it is negligible compared to screw length in that machine. From a functional perspective, in fact, the material feeding is performed by screw length instead of screw diameter considering the µIM machine screw design. Since the gap between the screw and the barrel is constant in the µIM machine, the diameter has less influence with respect to the length of the active screw. The volume fed can be actually calculated as a cross sectional area of the gap multiplied by the feeding length (that is the result of n rotations of the crew multiplied the screw pitch).

Differently from previous functional analysis, Table 8 and Table 9 also include “store (pellet)” function within functional requirements and notch screw diameter within design parameters. This phase was not considered in previous functional maps, but it is inserted in the following axiomatic matrix in order to obtain the complete matrix.

The two previous matrixes (Table 8 and Table 9) show that µIM machine presents an uncoupled design considering the axiomatic perspective, each machine function being carried out by a different design parameter (diagonal matrix). On the other hand, the IM machine presents a coupled design.

The entire axiomatic analysis finally confirms the µIM machine uncoupled design and the presence of “control functions” (illustrated in yellow in Table 9), only owned by the µIM machine design. The axiomatic design theory explains that a decouple design outperforms a coupled one: Actually, it means that no further optimization of the system is required since each parameter can be managed separately with respect to all others. For these reasons, new µIM design over-performs the standard design of a conventional IM machine.

### 3.2. Experimental Results

In order to understand what is the impact of the functional analysis and axiomatic design conclusions in terms of manufacturing precision and accuracy, the two batches produced by the IM and the µIM machines were compared.

The replication performance of IDb and ODt was evaluated by means of a shrinkage indicator S, which was defined as:(4)$$S\;=\;\frac{D_{\mathrm{polymer}}{\;-\;D}_{\mathrm{mould}}}{D_{\mathrm{mould}}}$$
where D_polymer_ and D_mould_ represent the same diameter measured on the moulded parts and the mould respectively. Such a variable allows evaluating the real shrinkage of the polymer, since the influence of the mould dimension is eliminated through the normalization.

#### 3.2.1. Comparison Based on Replication of Diameters

The results of IDb measurements are shown in Figure 14. What stands out is that µIM allowed to attain a better replication for all cavities, the S value always being closer to 0, i.e., the perfect replication of the mould feature. In fact, IDb shrank five times more when using IM with respect to µIM. This improvement was due to a more efficient filling phase in µIM resulting from the faster injection and to a more effective holding phase caused by faster switch-over and smaller injection volume (see Table 5 and related discussion). As for the different cavities, both the technologies resulted in a balanced multi-cavity replication process: The interval bars in fact overlap for the four cavities of IM and µIM. µIM also leads to a better repeatability of IDb, the interval bars always being smaller than for the IM case.

Figure 15 shows the results for the replication of ODt. µIM provided a generally better replication also in this case. However, the benefit of using the micro-scaled technology is not as evident as with IDb. In this case, in fact, a certain deviation between the different cavities was observed for both IM and µIM: Cavities 1 and 3 of IM were replicated with a level comparable to µIM, while cavity 2 and 4 provided a lower replication performance, proving that cavity unbalance still affected the outcome of the process. µIM also provided results that varied with the cavity, although less than the conventional technology.

When comparing the replication performances of the two diameters, it is possible to observe that the benefit introduced by using a micro-injection moulding machine was more pronounced for IDb. In fact, when considering ODt, there was not substantial improvement in terms of replication observed for IDb when using µIM instead of IM. Such a difference might be due to the fact that IDb was obtained by replicating a pin, while ODt by replicating an outer geometry. Thus, the polymer was free to shrink when generating ODt, but it was not in the case of IDb, since the presence of the pin did not allow a free contraction of the polymer. Such a constrained deformation generates a concentration of residual stresses in correspondence with internal geometries such as holes that increase the shrinkage of the moulded part once it is ejected from the cavity [23]. Since the shrinkage amount of IDb substantially decreased when applying µIM, it may be possible that the use of µIM instead of IM allowed reducing the residual stresses and the deriving shrinkage.

In order to compare the repeatability of IDb and ODt achieved with the two processes, the capability index Cp was used. Such a parameter allows to evaluate the variability of a process with respect to the imposed design specifications [29] and is calculated as:(5)$$C_{p}\;=\;\frac{\mathrm{USL}\;-\;\mathrm{LSL}}{6\sigma }$$
where USL and LSL are the upper and lower specification limits as set by the tolerance, and σ is the standard deviation of the results. A higher Cp is the result of a more precise, i.e., repeatable, process. In manufacturing productions, values larger than 1.33 are considered as satisfying as the process operates with a four-sigma performance. In the case of SIDb and SODt, USL and LSL were set by applying Equation (4) to the tolerance of 50 µm. The target, i.e., the mean between USL and LSL, was set at perfect replication, i.e., at a value of S equal to 0.

Figure 16 shows the distribution of SIDb values gathered from the four cavities. It can be seen that µIM yielded a more precise production, the data being less disperse than for IM. It is also clear how the micro-scaled technology manufactured parts have IDb much closer to the mould dimensions, as anticipated before. The Cp values for the two cases are reported in Table 8. For both processes, a Cp larger than 1.33 was attained. However, µIM proved much more repeatable with a Cp. of 5.11, confirming the results of the functional and axiomatic analyses in Section 3.1 and the related conclusion concerning the Im and µIM machines design differences.

The distributions of the results of S_ODt_ are shown in Figure 17. For this measurand, the two distributions overlapped, demonstrating that the data gathered with the two technologies were more comparable than in the previous case. However, a narrower distribution and consequently more precise production was once again attained with µIM, as also confirmed by the higher Cp value (see Table 10).

#### 3.2.2. Comparison Based on Density Results

The results of the density calculations are reported in Figure 18. What stands out is that the parts moulded using µIM had a significantly higher density, closer to the nominal value of the material data-sheet equal to 0.89 g/cm^3^. The reason for this lies in the more effective holding phase achieved with the µIM machine. In fact, the holding pressure acts on the value of the specific volume of the polymer melt. In particular, an effective application of the holding pressure yields an increase of the density of the final part by minimizing reduction of specific volume suffered by the polymer melt and the consequent shrinkage. In addition, the smaller feed system adopted in combination with the µIM machine allowed to postpone the freezing of the gate, thus leaving a wider window open for the action of the holding pressure. Furthermore, the functional analysis in Section 3.1 gives evidence of the smaller amount of air and to its easier evacuation from µIM injection chamber, supporting the aforementioned experimental results.

As for the dispersion of the results, comparing the distributions of the results provided by the two processes (see Figure 19) shows that IM parts had a more heterogeneous density. Therefore, the adoption of µIM was also beneficial with respect to repeatability. This was most probably caused by the enhanced precision of the µIM machine due in turn to its electric drives, lighter injection piston and more homogeneous polymer melt, which resulted in a more repeatable injection procedure.

## 4. Conclusions

The present paper aimed at comparing IM and µIM by means of functional analysis and experimental data when moulding the same micro TPE component. The functional analysis was used to identify the main differences between the conventional and the micro injection moulding machines that were used in the experimental campaign. In particular, each of the phases carried out by the machines (plastication, feeding, injection and packing) was analysed in terms of functionality by considering the different energies involved. The two machines were also compared by building axiomatic matrixes, which allowed assigning the functions to the main design parameters. The functional analysis allowed drawing the following conclusions:The friction generated by the IM machine, during the plastication phase, was greater than that of the µIM machine because of the bigger screw size and mass.The smaller dimension and tighter tolerances of the µIM machine screw make the metering procedure more precise and accurate and drastically reduce the backflow effect in the injection phase.The presence of the bored hole and pressure sensor in the µIM machine generates a closed loop control that is absent in IM.The µIM machine plunger, being lighter than the IM machine screw, allows to carry out a faster acceleration, thus obtaining a more effective injection phase.The IM screw begins the injection phase still while accelerating. Instead, the µIM plunger begins the injection phase after reaching the desired injection acceleration, thus allowing to obtain a more accurate and precise injection phase.In accordance to the functional analysis conclusions, the axiomatic design showed that the IM machine has a coupled design, whereas the µIM machine has an uncoupled one, thus allowing a higher controllability of each design parameter.The experimental observations allowed concluded that:Selecting the µIM process resulted in a great reduction of material waste, the feed system being much smaller than that adopted with IM.µIM provided a relevant replication improvement if compared to IM concerning the inner bottom diameter IDb of the moulded parts. The replication was also more precise, resulting on a higher Cp value.As for the outer top diameter ODt, µIM improved the replication accuracy and precision, even though it was less than the other measurand. This discrepancy could be due to the fact the IDb dimension is more influenced by the residual stress build-up being an inner geometry and as such, the subject of constrained shrinkage and less sensitive to the machine design and performance.The µIM process provided parts having a substantially higher and more homogeneous density among the four cavities. This clearly proved that the more repeatable injection phase and more effective packing phase allowed to better compensate the volumetric shrinkage of the polymer towards the end of the moulding process.

The results of this research unveiled the design reasons behind the higher performance of the µIM machine. This was obtained by the reverse engineering of the IM and µIM machines design and by establishing the links between theoretical analyses (functional analysis and axiomatic design) with experimental results. The interpretation of these results provides a valuable insight for both the production engineers and micro product designers in order, on one hand, to improve such a challenging manufacturing process as µIM, and on the other to develop new micro product designs that can take full advantage of the possibilities given by the µIM technology.

## Figures and Tables

**Figure 1 micromachines-11-01115-f001:**
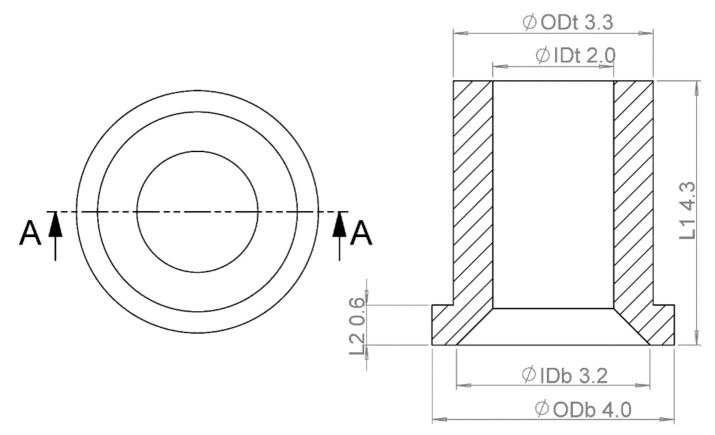
Micro part with main geometries indicated. The nominal dimensions in mm are shown.

**Figure 2 micromachines-11-01115-f002:**
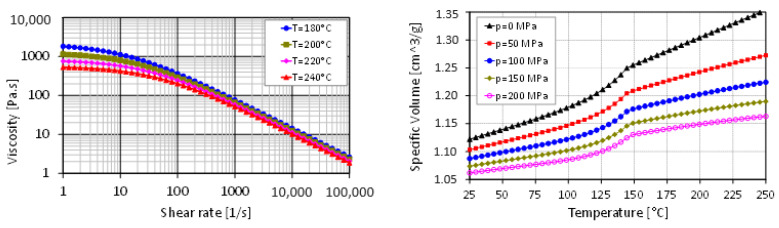
Viscosity plot (Left) and pressure-specific volume-temperature plot (Right) of the thermoplastic elastomer material used in the injection and micro injection moulding experiments.

**Figure 3 micromachines-11-01115-f003:**
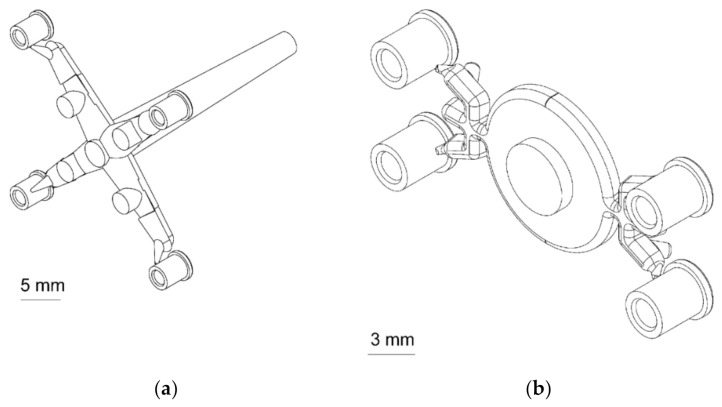
3D view of the feed systems used with (**a**) the IM machine and (**b**) the µIM machine. Reproduced with permission from [23].

**Figure 4 micromachines-11-01115-f004:**
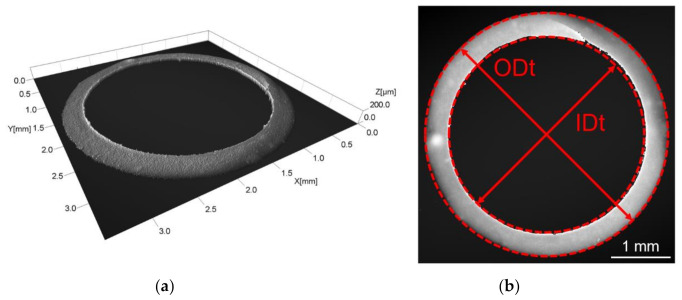
Measurement performed with the focus variation microscope: (**a**) 3D acquisition of the top of the micro part reproduced with permission from [23]; (**b**) measurement of ODt and IDt by fitting the corresponding circles.

**Figure 5 micromachines-11-01115-f005:**
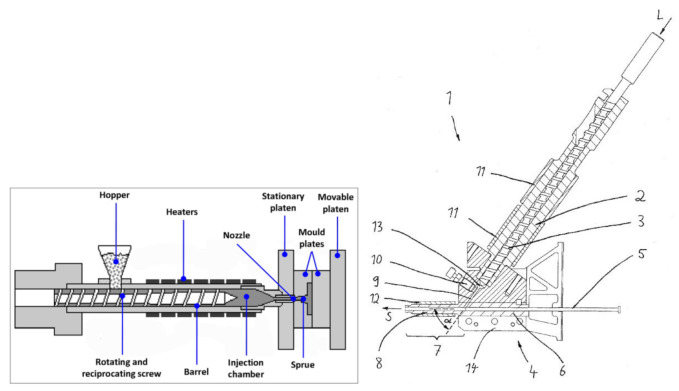
Elements of a conventional injection moulding machine (**left**). Reproduced with permission from [17]. Elements of the micro injection moulding machine considered in the present study (**right**): (1) plasticizing and injection unit, (2) screw cylinder, (3) plasticizing cylinder, (4) injection element, (5) injection piston, (6) injection cylinder, (7) metering section of the injection cylinder, (8) injection nozzle, (9) flow path, (10) pressure sensor, (11) heating element, (12) heating element, (13) volume, (14) distributor lock, (L) longitudinal axis, (S) injection cylinder axis, (α) angle between the screw and the injection cylinders.

**Figure 6 micromachines-11-01115-f006:**
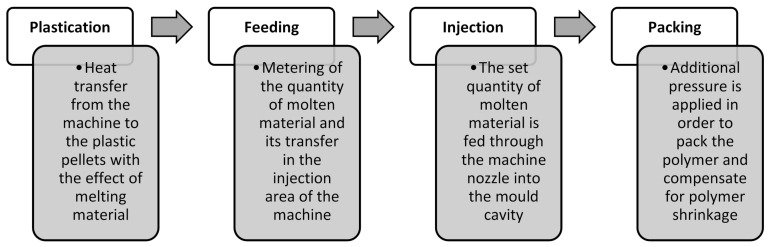
The phases of the moulding process performed by an injection moulding machine.

**Figure 7 micromachines-11-01115-f007:**
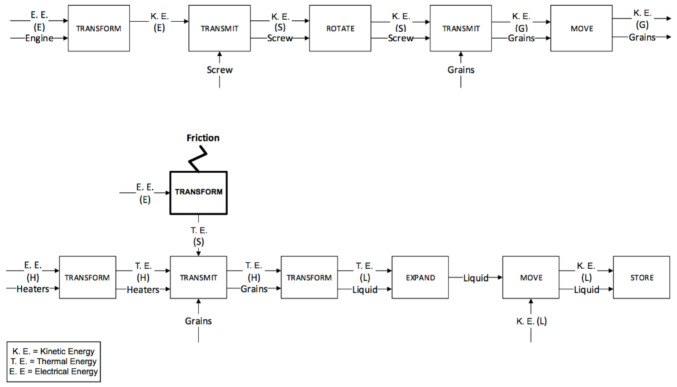
Plastication phase for the IM machine.

**Figure 8 micromachines-11-01115-f008:**
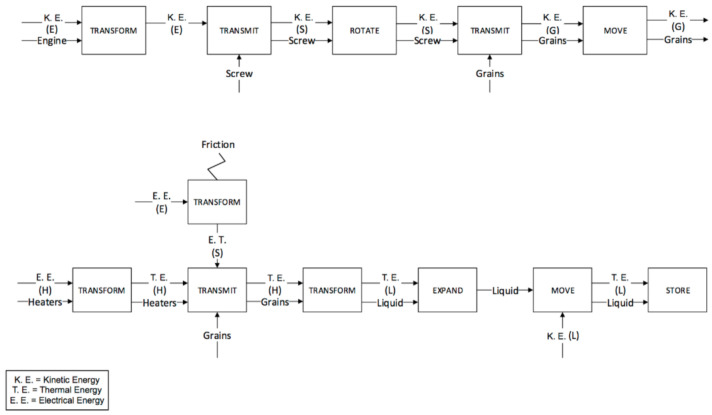
Plastication phase for the µIM machine.

**Figure 9 micromachines-11-01115-f009:**
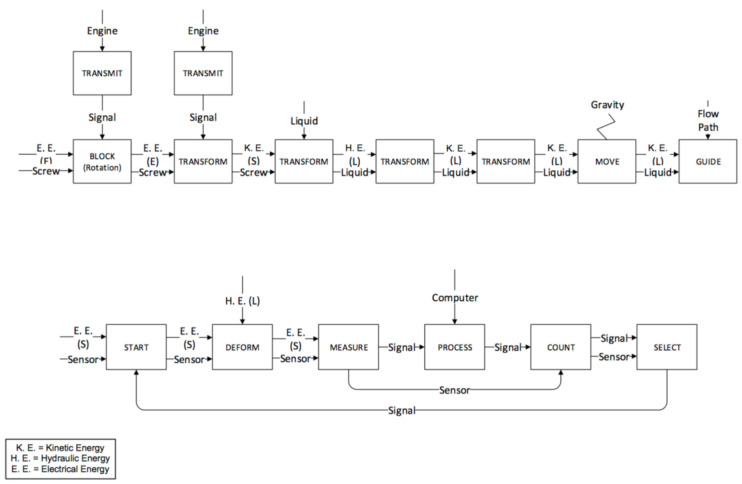
Feeding phase for the µIM machine.

**Figure 10 micromachines-11-01115-f010:**
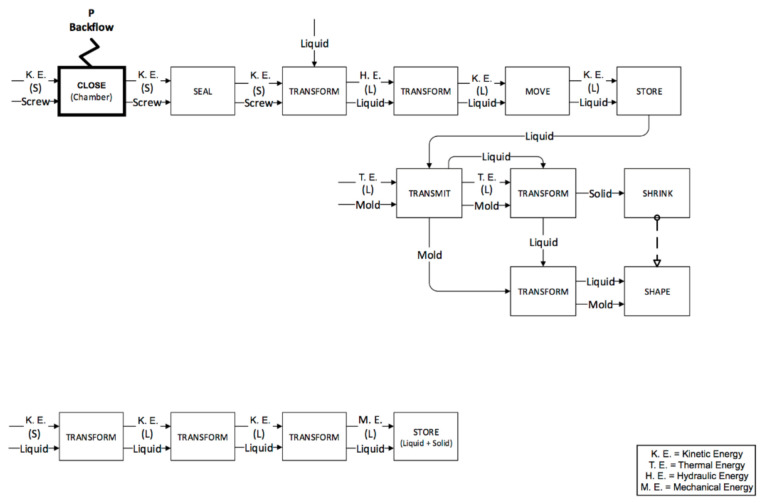
Injection phase for the IM machine.

**Figure 11 micromachines-11-01115-f011:**
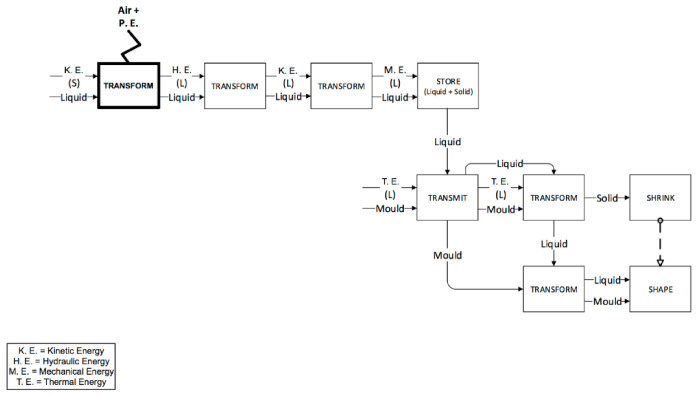
Packing phase for the IM machine.

**Figure 12 micromachines-11-01115-f012:**
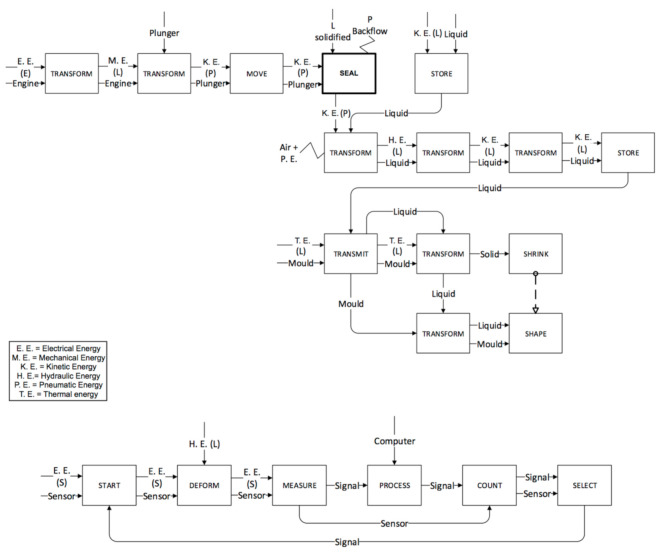
Injection and packing phases for the µIM machine.

**Figure 13 micromachines-11-01115-f013:**
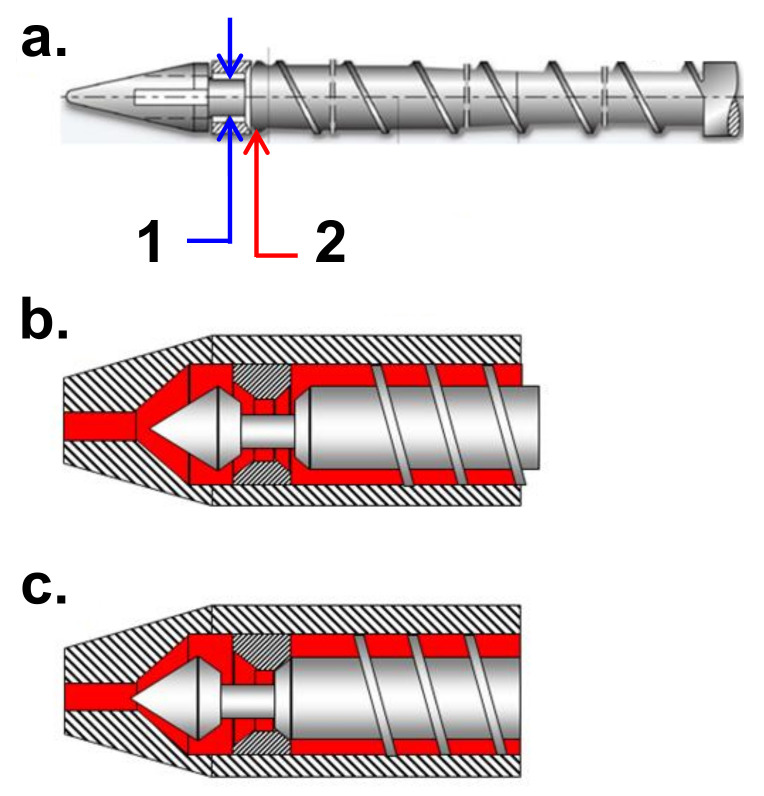
(**a**) IM screw features: (1) diameter of the restriction of the injection chamber and valve (2); features (1) and (2) positions during (**b**) feeding and (**c**) injection. Reproduced with permission from [28].

**Figure 14 micromachines-11-01115-f014:**
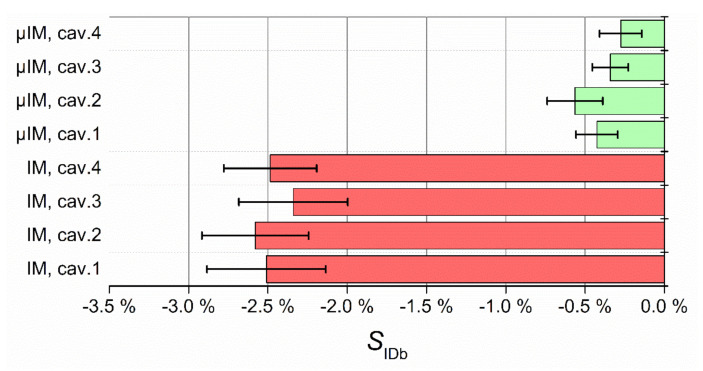
Replication results for IDb. The interval bars represent the 95% confidence intervals for the mean.

**Figure 15 micromachines-11-01115-f015:**
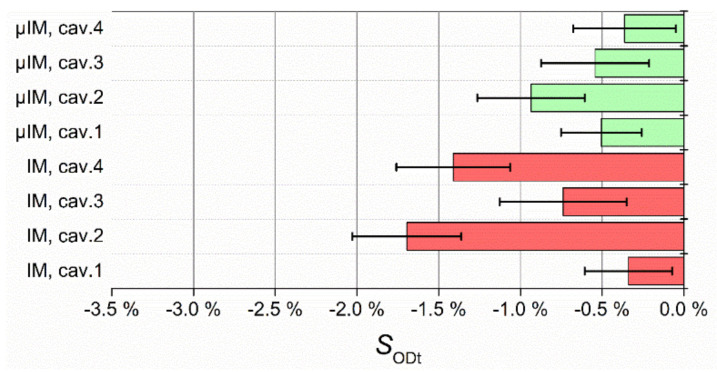
Replication results for ODt. The interval bars represent the 95% confidence intervals for the mean.

**Figure 16 micromachines-11-01115-f016:**
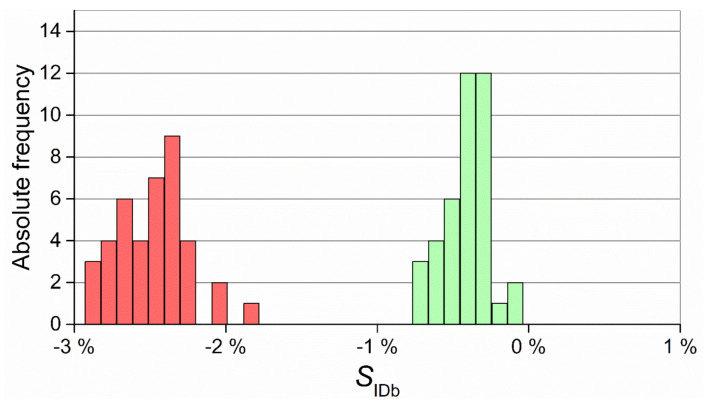
Distribution of SIDb values for IM (red) and µIM (green).

**Figure 17 micromachines-11-01115-f017:**
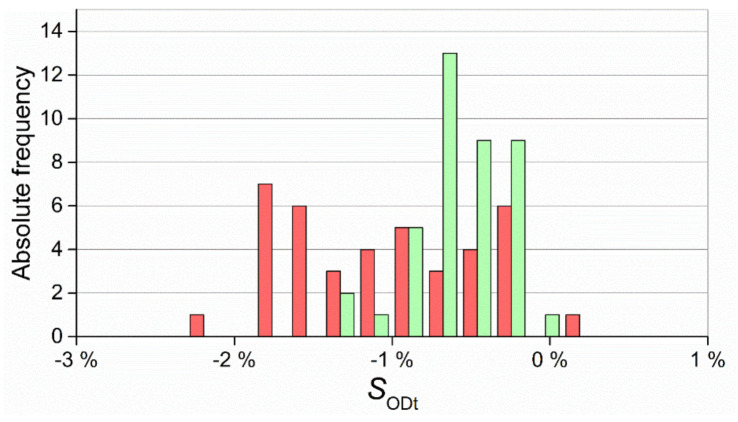
Distribution of SODt values for IM (red) and µIM (green).

**Figure 18 micromachines-11-01115-f018:**
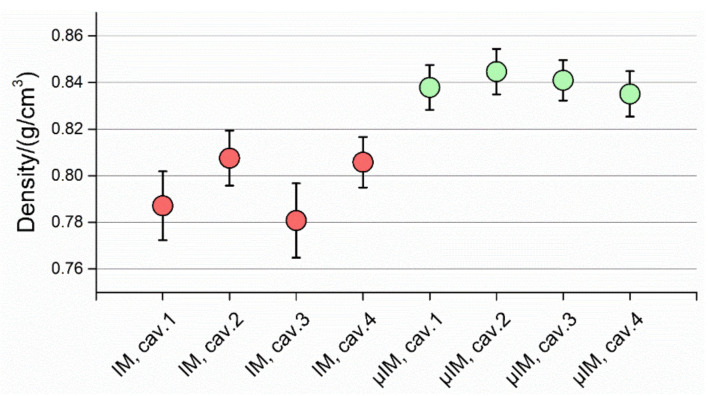
Density calculations for IM (light blue) and µIM (red). The interval bars indicate the standard deviations of the 10 parts per cavity.

**Figure 19 micromachines-11-01115-f019:**
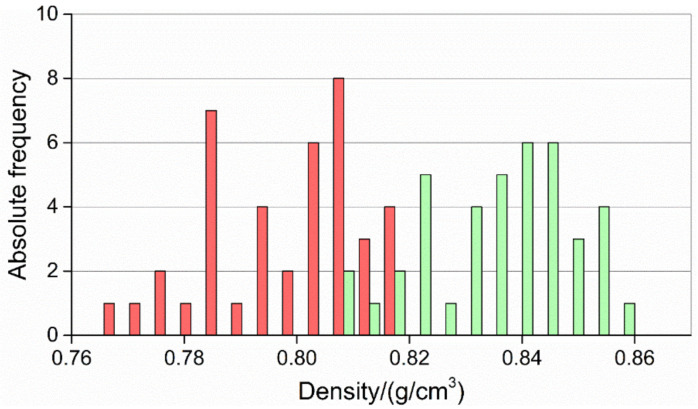
Distribution of density values for IM (red) and µIM (green).

**Table 1 micromachines-11-01115-t001:** Process parameters for IM and µIM, Reproduced with permission from [23].

Process Parameter	IM	µIM
Injection speed/(mm/s)	40	160
Holding pressure/bar	350	350
Melt temperature/°C	220	220
Mould temperature/°C	40	40
Cycle time/s	17	8

**Table 2 micromachines-11-01115-t002:** Average uncertainty contributions and expanded uncertainty U for the IM parts.

Uncertainty Contribution	IDb/µm	ODb/µm	IDt/µm	ODt/µm	L1/µm	L2/µm
u_cal_	0.50	0.50	0.50	0.50	0.35	0.35
u_p_	0.09	0.09	0.09	0.09	0.17	0.17
u_w_	0.54	0.59	0.63	0.36	1.01	1.00
u_res_	0.51	0.51	0.51	0.51	0.14	0.14
U (k = 2)	1.8	1.9	1.9	1.6	2.2	2.2

**Table 3 micromachines-11-01115-t003:** Average uncertainty contributions and expanded uncertainty U for the µIM parts.

Uncertainty Contribution	IDb/µm	ODb/µm	IDt/µm	ODt/µm	L1/µm	L2/µm
u_cal_	0.50	0.50	0.50	0.50	0.35	0.35
u_p_	0.09	0.09	0.09	0.09	0.17	0.17
u_w_	0.38	0.53	0.45	0.85	0.81	1.2
u_res_	0.51	0.51	0.51	0.51	0.14	0.14
U (k = 2)	1.6	1.8	1.7	2.2	1.8	2.5

**Table 4 micromachines-11-01115-t004:** Feature blocks of the metering phase for the two machines.

Process	D_screw_	T_screw_	d_screw_	t_screw_	L_screw_	l_screw_	Pressure Sensor	D Bored Hole
IM machine	X	X			X			
µIM machine			X	X		X	X	X

**Table 5 micromachines-11-01115-t005:** Feature blocks of the injection phase for the two machines.

Process	D_screw_	T_screw_	d_plunger_	t_plunger_	L_screw_	L_plunger_	Strain Gauge Sensor
IM machine	X	X			X		
µIM machine			X	X		X	X

**Table 6 micromachines-11-01115-t006:** List of design parameters of IM machine.

Design Parameter	Description
D notch_screw_	Diameter of ideal cylinder tangent to thread end
D_screw_	Screw nominal diameter
N of turns	Number of screw turns during the feeding phase
T_screw_	Screw tolerance
L_screw_	Screw length
Stroke (l)	Length between the screw starting and end point within the injection chamber

**Table 7 micromachines-11-01115-t007:** List of design parameters of µIM machine.

Design Parameter	Description
d notch_screw_	Diameter of ideal cylinder tangent to thread end
N of turns	Number of screw turns during the feeding phase
l_screw_	Screw length
Def. Pressure Sensor	Sensor deformation entity which corresponds to the measure controlled by the sensor (Young modulus plus sensor geometry)
d_plunger_	Plunger diameter
Stroke (l)	Length between the plunger starting and end point within the injection chamber
D bored hole	Diameter of the bored flow path (9 in Figure 5)
t_plunger_	Plunger diameter tolerance
Def. Sensor S.G.	Sensor deformation entity which corresponds to the measure controlled by the sensor (Young modulus plus sensor geometry)

**Table 8 micromachines-11-01115-t008:** Axiomatic matrix for the IM machine.

Machine Design Parameter	D _notchscrew_	D_screw_	N of Turns	T_screw_	L_screw_	Stroke (l)
Store (pellets) (feeding phase)	X					
Feed (feeding phase)	X	X	X			
Seal (feeding phase)				X	X	
Store (feeding phase)		X				X
Meter (feeding phase)	X	X	X			
Move (injection phase)						X
Seal (injection phase)						X

**Table 9 micromachines-11-01115-t009:** Axiomatic matrix for the µIM machine. Control functions are indicated in yellow.

Heading	d notchscrew	N of Turns	lscrew	Def. Pressure Sensor	dplunger	Stroke (l)	D Bored Hole	tplunger	Def. Sensor S.G.
Store (pellets) (feeding phase)	X								
Feed (feeding phase)	X	X							
Seal (feeding phase)			X						
Meter (feeding phase)				X					
Store (feeding phase)					X	X			
Set (feeding phase)							X		
Seal (injection phase)								X	
Move (injection phase)						X			
Measure (injection phase)									X

**Table 10 micromachines-11-01115-t010:** Axiomatic matrix for the µIM machine. Control functions are indicated in yellow.

Cp	IM	µIM
SIDb	2.11	5.06
SODt	1.85	2.11

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
