# Peer review of "Functional Analysis Validation of Micro and Conventional Injection Molding Machines Performances Based on Process Precision and Accuracy for Micro Manufacturing"

_micromachines, 2020, doi:10.3390/mi11121115_

Round 1

Reviewer 1 Report

The authors correlate the IM and μIM machine architectures' functionality to the actual dimensional capabilities of both the macro and micro-processes.

The article is well written, and the results support conclusions.

The abstract is not very clear and should be rewritten.

Reviewer 2 Report

“Functional analysis validation of micro and conventional injection molding machines performances based on process precision and accuracy for micro manufacturing” analyses the performances of two technologies, injection moulding and micro-injection moulding to manufacture micro  polymer  parts  and evaluates which technology is more suitable to reach the accuracy and precision required by micro products.

Revision or comment

The work is interesting and evaluates quantitatively the performances of the two technologies in micro part manufacturing, with a good analyses on the functional aspects involved in the two “different” approaches. However, in particular, two aspects should be considered. It is not evident if the functional analyses results could be extended to other µIM machines and the chosen moulded component is not properly a micro part according to conventional definition (Whiteside et al. (2003) Micromoulding: process characteristics and product properties. Plast Rubber Compos 32:231–239).

Introduction is adequate to the issue but should be comprehensive of a wide field and experience present in literature.
